



# Massive corals record deforestation in Malaysian Borneo through sediments in river discharge

Walid Naciri [1], Arnoud Boom [1], Matthew Payne [1], Nicola Browne [3], Noreen J. Evans [2], Philip Holdship [4], Kai Rankenburg [2], Ramasamy Nagarajan [5,6], Bradley J. McDonald [2], Jennifer McIlwain [3], Jens Zinke [1,3]

[1] School of Geography, Geology and the Environment, University of Leicester, Leicester, LE1 7RH, United Kingdom

[2] School of Earth and Planetary Sciences / John de Laeter Centre, Curtin University, Bentley, WA 6102, Australia

[3] Molecular and Life Sciences, Curtin University, Bentley, WA 6102, Australia

[4] School of Earth Sciences, Oxford University, Oxford, OX1 2JD, United Kingdom

[5] Department of Applied Sciences (Applied Geology), Curtin University Malaysia, Miri, 98009, Malaysia

[6] Curtin Malaysia Research Institute, Curtin University Malaysia, Miri, 98009, Malaysia

*Correspondence to*: Walid Naciri (wn36@leicester.ac.uk)

**Abstract.** Logging of tropical primary forests is a widely acknowledged global issue threatening biodiversity hotspots and indigenous communities leading to significant land erosion and decreased soil stability. The downstream effects of logging on human coastal communities include poor water quality and increased sedimentation. Quantifying the impacts of historical deforestation within a watershed requires accurate data from river discharge or satellite images, which are rarely available prior to the 1980's. In the absence of these in-situ measurements, proxies have successfully produced accurate, long-range, historical records of temperature, hydrological balance, and sediment discharge in coastal and oceanic environments. We present a 30 year, monthly resolved Ba/Ca proxy record of sediment in river discharge as measured from the skeletal remains of massive corals *Porites* sp. from northern Malaysian Borneo. We make the comparison with local instrumental hydrology data, river discharge and rainfall, to test the reliability of the Ba/Ca$_{coral}$ proxy. Our results show that averaging five records into two composites results in significant positive correlations with river discharge (r = 0.5 and r = 0.59) as well as a difference in correlations strength coherent with distance from the river mouth, with the composite closer to the river mouth displaying a higher correlation. More importantly, *Porites* sp. corals from this region recorded a very similar upward trend to that of river discharge on multi decennial time scales. The lack of similar increase, and overall stability in the precipitation record suggests that the river discharge's trend recorded by corals is linked to the increasing land use associated with ever–growing deforestation. We argue that massive corals in this region are therefore valuable archives of past hydrological conditions and accurately reflect changes in land use patterns.

## 1 Introduction

Southeast Asia hosts the most extensive and diverse coral reefs in the world, located in the Coral Triangle within the Maritime Continent (Burke et al., 2011). However, it has been estimated that the consequences of human population growth, coastal development, overfishing, trawling and pollution negatively impact 95 % of coral reefs in the Southeast Asia region, resulting in coral cover and diversity declines (Burke et al., 2011). This decline is likely due to the combination of larger-scale threats such as climate change (Hoegh-Guldberg et al., 2007), as well as increased land use and deforestation since the middle of the 20[th] century to satisfy the global demand for oil palm and pulpwood (Pittman and Carlson, 2013; Miettinen et al., 2016; Gaveau et al., 2019).



Estimates show that up to 50 % of the world's coral reefs are under threat from terrestrial runoff in areas under intense land–use change such as Borneo and across SE Asia (Burke et al., 2011). The island of Borneo, which includes Kalimantan (Indonesia)

and Sarawak (Malaysia), was historically covered by dense rainforest. However, as of 2015, 34 % of its old-growth forest coverage has been lost since 1973 (Gaveau et al., 2016, 2019) with up to 60 % of said lost surface area rapidly converted into plantations (Gaveau et al., 2016) (Fig. S1). Forest clearance has been shown to increase soil erosion (McDonald et al., 2002; Sidle et al., 2006), which in turn enhances suspended sediment fractions transported by rivers (McCulloch et al., 2003; Lewis et al., 2007; Fleitmann et al., 2007). This riverine suspended sediment fraction carries a chemical signature of the catchment rocks

and soils, modified by human activity (e.g. industry, sewage, dams etc.) (Martin et al., 2018; Prabakaran et al., 2020; Lihan et al., 2021; Liong et al., 2021). Enhanced sediment delivery to rivers and coasts results in decreased light availability through increased turbidity and light attenuation (Storlazzi et al., 2015). Low light, suspended sediments, and sediment deposition have a negative effect on coral reefs which is well documented. The effects include: reduced coral growth (Bessell-Browne et al., 2017), increased bioerosion (Chen et al., 2013), increased coral mortality through coral smothering and burial (Rogers, 1990; Weber et

al., 2012), decreased coral recruitment as well as diversity (Fabricius et al., 2003; Fabricius, 2005) and increased coral disease (MacNeil et al., 2019). Coral reefs exposed to enhanced sediment delivery may be more resistant to bleaching events due to thermal stress (Glynn, 1996; Browne et al., 2019), however, their recovery after a disturbance is slower than corals exposed to a higher water quality in addition to the many drawbacks stated before (MacNeil et al., 2019). Hence, the cumulative effects of these local and global stressors such as warming oceans, ocean acidification, and sea level rise, could lower the resilience of

reefs to future disturbances (Carilli et al., 2009).





Figure 1: (a) Deforestation extent from 2001 to 2019 across the Baram River basin (map created on Python with annual deforestation data derived from the Global Forest Change dataset (Hansen et al., 2013)), © OpenStreetMap contributors 2022. Distributed under the Open Data Commons Open Database License (ODbL) v1.0", (b) annual deforestation rate in the Baram River catchment in hectares between 2001 and 2019 and (c) map of the South China Sea and surrounding countries with an enlarged map of the study area both showing the bathymetry as well as the two sampling sites, Eve's and Anemone's Garden, the city of Miri, Miri River, and the Marudi station along the Baram River on the focused map. Map made on QGIS® (v. 3.22.0-Białowieża) using raster data from Natural Earth.





Previous studies have shown the Baram River catchment, mainly formed of shale, sandstone, and minor amounts of limestone (Vijith and Dodge-Wan, 2018) to be prone to land erosion because of deforestation, with heavy logging areas creating erosion hotspots (Vijith et al., 2018a). Although 64 % of the catchment shows a low erosion risk, logging–induced barren land (4 % of the area) accounts for 28 % of soil loss (Vijith et al., 2018a, b), stressing the disproportionate effect of deforestation.

Furthermore, Browne et al. (2019) showed that reefs in the Miri–Sibuti Coral Reef National Park were subjected to high sediment load and high bioerosion that gradually lessened as the distance separating the coral reefs and river mouth increased. The largest impact is thought to derive from the Baram River discharging $2.4 \times 10^{10}$ kg yr$^{-1}$ of sediment (some barium rich) into the coastal environment (Prabakaran et al., 2019), most likely in response to changes in land use and human development (Straub and Mohrig, 2009; Nagarajan et al., 2015). However, long term records of sediment delivery into the coastal ocean are currently

lacking.

Historic environmental conditions influence how coral reefs respond to future disturbance events. Therefore, understanding what the environmental conditions were and how corals on reefs responded to these conditions improves our ability to assess future reef trajectories. Unfortunately, the assessment of climatic and environmental change in the tropics affecting terrestrial and

coastal marine environments is hampered by the lack of long-term weather or river gauging stations as well as long-term water quality monitoring. While accurate information on regional climate can be obtained from historical ship-based observations (e.g. ICOADS historical sea surface temperature data set (Parker et al., 1995)), others only have precise regional scale environmental data, which has been globally available since aerial observation and, more importantly, satellites imaging, commenced in the early 1980s (e.g. Landsat images for forest clearance; land surface temperature and rainfall). River discharge data for tropical

catchments are extremely scarce or short in duration, covering 20–30 years or more in rare cases (Sa'adi et al., 2017b), and suspended sediment concentrations and other water quality measures in river runoff are even more scarce (Syvitski et al., 2000). This lack of data hinders the study of temporal and spatial variability of natural change and anthropogenic impacts on river catchment hydrology as well as their effects on downstream ecosystems. Consequently, to assess historical land use change and its impact on terrestrial and coastal marine environments, biological proxy archives such as corals and molluscs have been used

to expand the environmental record (Schöne, 2013; Saha et al., 2016).

Corals such as *Porites* sp. form boulder–like structures that can be sampled using a coring device and can be analysed for geochemical composition with high temporal resolution determined by coral growth rate (Saha et al., 2016; Thompson, 2022).

During their growth, these corals can integrate trace elements present in the surrounding seawater into their calcium carbonate skeleton in proportion to their concentration in seawater (Saha et al., 2016; Thompson, 2022). Based on the same principle, the skeleton's stable oxygen isotope ($\delta^{18}$O) signature is driven by a combination of the sea surface temperature (SST) and the surrounding seawater $\delta^{18}$O signature, which, depends on the evaporation/precipitation balance and river runoff into tropical oceans (Ren et al., 2003; Cahyarini et al., 2008; Reed et al., 2022).

Although, vital effects such as uneven calcification rate can disrupt the absorption of trace elements proxies such as Sr/Ca, due to its inverse correlation with calcification rate (Grove et al., 2012; Kuffner et al., 2012). The presence of secondary aragonite following post-depositional diagenesis can lead to unusually enriched $\delta^{18}$O and higher Sr/Ca values (Quinn and Taylor, 2006). However, both vital effects and diagenesis are relatively easy to circumvent in modern coral records (Thompson, 2022). As such, massive corals provide a living archive of past environmental conditions with precise internal chronologies (Weber and

Woodhead, 1972; Sinclair et al., 2006; Thompson, 2022). Further, as these species can live for several hundred years, they often provide a considerable extension to the instrumental data record (Tierney et al., 2015; Zinke et al., 2022).

Coral skeletons can also be used to create historical reconstructions of sea surface salinity (SSS) as well as sediment discharge. Changes in SSS are obtained from reconstructions of the seawater's $\delta^{18}$O composition ($\delta^{18}$O$_{sw}$) derived from coupling coral

skeletal Sr/Ca, the most robust sea surface temperature proxy (de Villiers et al., 1995; Corrège, 2006; DeLong et al., 2013) and $\delta^{18}$O (McCulloch et al., 1994; Ren et al., 2003; Cahyarini et al., 2008). A recent study revealed that records of $\delta^{18}$O$_{sw}$ derived from corals such as *Porites* sp. in the Miri–Sibuti Coral Reef National Park, were able to provide reliable information on the changes in the hydrological balance in the region, and showed significant correlation with river discharge (Krawczyk et al., 2020). Additionally, these corals recorded a decreasing trend in $\delta^{18}$O$_{sw}$ values between 1982 and 2016, corresponding to a

freshening of ambient seawater (Krawczyk et al., 2020).

Ba/Ca and Y/Ca as well as rare earth elements have been used to reconstruct sediment discharge in river water transported to the coastal zone (Moyer et al., 2012). Several studies on cross–shelf gradients in coral reefs showed how coral colonies in proximity



to river outputs have high rates of disturbance from sedimentation as indicated by the coral Ba/Ca content (Jupiter et al., 2008; Moyer et al., 2012; Grove et al., 2012; Brenner et al., 2017; Chen et al., 2020; D'Olivo and McCulloch, 2022). Coral Ba/Ca
relies on the fact that the Ba in seawater originated primarily from terrestrial soil suspended in river runoff and is therefore an indicator of sediment input into the marine environment (McCulloch et al., 2003; Lewis et al., 2007; Fleitmann et al., 2007).

In this study we aim to expand on earlier findings regarding changes in SST and $\delta^{18}O_{sw}$ in the Miri–Sibuti Coral Reef National Park by Krawczyk et al. (2020) by developing the first record of Ba/Ca ratios in multiple *Porites*' skeletons as a proxy for
sediment in river discharge between 1985 and 2015 (Moyer et al., 2012; Brenner et al., 2017; Chen et al., 2020; Grove et al., 2012). These new Ba/Ca records will establish if land erosion associated with increasing land use and deforestation in the Sarawak region has reached corals in the coastal area through high sediment input in riverine waters and thus define the level of connectivity between the watershed and coral reefs (Gaveau et al., 2014; Vijith et al., 2018b).
We test the hypothesis that coral Ba/Ca records from our study site can accurately record sediment in river discharge changes
and therefore changes in land use in the Baram River catchment. This assessment will provide novel data and confirm if corals in the Miri–Sibuti Coral Reef National Park has been exposed to increasing sediment loads since deforestation commenced, thus providing important evidence for future environmental land and coastal management policies.

## 2 Methods


### 2.1 Regional setting and climate

The South China Sea region is part of the Maritime Continent that separates the Indian and Pacific Oceans. Regional hydroclimate is mainly influenced by the East Asian Monsoon which can be divided into two monsoon seasons; a dry and warm summer monsoon that begins with the onset of south–westerly winds in late June, and a wet and colder winter monsoon starting
with the onset of north–easterly winds in November. Both seasons are accompanied by two inter–monsoon seasons in April and October (Stephens and Rose, 2005; Tangang et al., 2012; Sa'adi et al., 2017b). Additionally, several climate phenomena influence the local climate (i.e. temperature, rainfall, and river runoff), although to a lesser degree (Gomyo and Kuraji, 2009; Yan et al., 2015; Sa'adi et al., 2017b; Pan et al., 2018). These include the El Niño–Southern Oscillation (ENSO) creating dry (wet) conditions during El Niño (La Niña) (Murphy, 2006; Tangang et al., 2012), the Indian Ocean dipole creating cooler
(warmer) conditions during positive (negative) phases and the Pacific Decadal Oscillation leading to colder (warmer) sea surface temperatures during the negative (positive) phase (Screen and Francis, 2016). The warm south–western winds in boreal summer (June to September) and the colder north–eastern winds in boreal winter (December to March) control most of the seasonal SST and rainfall variations (Stephens and Rose, 2005; Tangang et al., 2012; Sa'adi et al., 2017a).
Although seasonality in SST and rainfall is quite low in tropical SE Asia compared to subtropical settings, this region still shows
a 2.5°C range between the hottest (June) and coldest (February) months while rainfall shows an amplitude of 215 mm between the wettest and driest months despite high standard deviations (SD) in the rainfall record. River discharge from both the Baram and Miri Rivers transport freshwater and sediments to the nearshore coral reefs. However, river discharge data are only available for the larger Baram River (Sa'adi et al., 2017b) with August usually showing the lowest discharge (2072 m$^3$ s$^{-1}$) and November, December, and January show the highest with 2.55, 3.13 and 3.08 x10$^3$ m$^3$ s$^{-1}$, respectively (Sa'adi et al., 2017b).


### 2.2 Climate and land use data

We extracted SST data from the NOAA 0.25° daily Optimum Interpolation Sea Surface Temperature version 2 (OISSTv2) (Reynolds et al., 2007; Banzon et al., 2016), which combines satellite (Advanced Very High–Resolution Radiometer AVHRR),
buoys, and ships of opportunity data. In this dataset, missing data are interpolated to create a continuous record of the 0.25° resolved grid used in this study: 4.25–4.5 °N, 113.75–114.00 °E. These data originate from NOAA's NCEI and were downloaded from NOAA PSL's website (Home: NOAA Physical Sciences Laboratory, 2022).



We obtained local precipitation data for the Miri Airport station (WMO station No. 964490, 4.4° N, 114.0° E, elevation: 51 m a.s.l) from the Global Historical Climatology Network Monthly (GHCN–M version 2) quality–controlled–dataset, extracted from NOAA's NCEI and downloaded from the KNMI Climate Explorer platform (Trouet and Van Oldenborgh, 2013).

We obtained river discharge data for the Baram River watershed from the Department of Irrigation and Drainage of Malaysia. Although the small Miri River is believed to also influence our study site (if only marginally), only the Baram River has a record of discharge data. We used the Marudi station (4.17° N, 114.31 °E) discharge data because it provides the most accurate data
from a station closest to the Baram River mouth.

We obtained the EN4.2.0 (hereafter EN4) subsurface ocean temperature and salinity compiled datasets from the Hadley Centre. EN4 is interpolated at 1° grid resolution, we extracted the original data created by the Met Office Hadley Centre for the study site (4–5° N, 113.5–114.5° E) from the KNMI Climate Explorer platform (Trouet and Van Oldenborgh, 2013).

We created an annual deforestation time series spanning 2001–2019 (Figs. 1a, b) from the Global Forest Change dataset from
Hansen et al. (2013). We converted the time series to a projection suitable for Sarawak (UTM 49 N, WGS 1984) for accurate geometric analysis by clipping it to the boundary of the Baram River basin (Figs. 1a, b). Any statistics produced therefore reflected only deforestation within the Baram catchment. We created the deforestation map in Python (version 3.9) within the PyCharm integrated development environment (JetBrains, 2022).


### 2.3 Sample collection and treatment

We sampled coral cores from Eve's Garden (EG) and Anemone's Garden (AG), two sites off Miri's coast in the Miri–Sibuti Coral Reefs National Park in Sarawak, Malaysian Borneo. Both sites are located along an inshore to offshore depth gradient (Fig.
1c), with EG (4° 20' 36.0492" N, 113° 53' 53.9412" E) situated at 5 m depth and 28.3 km away from the Baram River's mouth while AG (4° 17' 31.8084" N, 113° 49' 33.2796" E) was situated south–east of EG, 36.4 km away from the river mouth within 8 m depth (with depth measured from the colony's top). We drilled the cores from massive *Porites* sp. corals using a SCUBA tank–driven pneumatic drill (Silverline air drill reversible) equipped with a diamond coated drill head.

We collected a total of five coral cores, three from EG and two from AG in August 2016 and May 2017. Core samples ranged
from 35 to 125 cm in length. We sectioned these cores longitudinally into 7-8 mm thick and 4 cm wide slabs along the main growth axis using a diamond blade precision saw. We then X–rayed the resulting slabs to assess the optimal sampling path closest to the main growth axis. We cut the core sections again for laser ablation ICP/MS analysis along the chosen sampling path to fit the required size in the sample cell (2.5 cm wide and 10 cm long). For the samples' cleaning, we followed a pre–existing approach (Nagtegaal et al., 2012). We pre–cleaned samples for 24 hours in a reagent grade sodium hypochlorite solution
NaClO (with 6–14 % active chlorine) bath diluted to a 1:1 ratio with deionised water (15 M cm⁻¹). We rinsed each slab three times with deionised water (15 M cm⁻¹) in an ultrasonic bath for 10 minutes at room temperature, and changed the deionised water between each run. Finally, we transferred the slabs in an oven to dry at 50 °C for 48 hours.

We collected seawater samples in October 2019 across a transect between the mouth of the Miri River and AG. This transect
encompassed eight different stations, AG, EG and six stations between EG and the river mouth (S1 to S6, each one separated by 2 km), all taken as duplicates. To ensure no contamination, we cleaned all HDPE 60 mL plastic vials according to the sampling and sample-handling protocols for GEOTRACES cruises (Cutter et al., 2017). We stored samples in the dark in a fridge at 4 °C.

### 2.4 Ba/Ca measurements

Laser ablation inductively coupled plasma mass spectrometry (LA–ICP/MS) was performed at the GeoHistory Facility in the John de Laeter Centre, Curtin University, Perth, Australia. Coral slabs were ablated using a RESOlution–SE 193 nm excimer laser equipped with a large format Laurin Technic S155 sample cell typically holding 3–4 coral slabs up to 10 cm in length, along with NIST 610/612/614 (Hollocher and Ruiz, 1995; Jochum et al., 2005) and MACS–3 (Wilson et al., 2008) standard
reference materials. Laser fluence was calibrated above the sample cell using a hand–held energy meter, and subsequent analyses



were performed in constant energy mode. The cell was flushed by ultrahigh purity He (320 mL min$^{-1}$) and N$_2$ (1.2 mL min$^{-1}$). High purity Argon was used as the ICP/MS carrier gas (~1 L min$^{-1}$). Standards and samples were ablated in line scans using an adjustable, rotating rectangular slit aperture set to 325 x 50 microns (width x length). To additionally clean the sample surfaces, a pre–ablation run was performed at 10 Hz laser repetition rate and 50 % spot overlap before the ICP/MS was connected to the

laser cell. Sample ablation and data acquisition was then performed with 20 µm cm$^{-1}$ scan speed, 10 Hz laser repetition rate, and on–sample laser energy of 3.2 J cm$^{-2}$. Individual coral slabs were ablated in single continuous runs of up to 90 minutes, bracketed by shorter ablations (~2 min) of the reference materials using identical laser parameters.

We performed all measurements using an Agilent 7700 quadrupole ICP/MS. Each analytical session consisted of initial gas flow and ICP–MS ion lens tuning for sensitivity and robust plasma conditions ($^{238}$U/$^{232}$Th ~1; $^{206}$Pb/$^{238}$U ~ 0.2; and $^{238}$UO/$^{238}$U

<0.004). Pulse–analogue (P/A) conversion factors were determined on the NIST 610 reference glass by varying laser spot sizes and/or laser repetition rate to yield 1–2 Mcps per element. For data acquisition, $^7$Li, $^{11}$B, $^{25}$Mg, $^{43}$Ca, $^{55}$Mn, $^{86}$Sr, $^{89}$Y, $^{137}$Ba, $^{208}$Pb, and $^{238}$U were collected with dwell times of 20 ms each after 40 s of baseline acquisition. The time–resolved mass spectra were then reduced using the 'Trace Elements' data reduction scheme in Iolite 4.3 (Jochum et al., 2005; Wilson et al., 2008; Woodhead et al., 2007; Paton et al., 2011). Whereas the primary reference materials used in this study for the correction of instrumental drift

and determination of elemental concentrations were homogeneous silicate glasses NIST 610/612 for AG1–2, AG5, and EG15 cores, and NIST 614 for EG3 and EG4, final trace element concentrations and element/Ca ratios were additionally normalized to the matrix–matched MACS–3 pressed carbonate reference material (S. Wilson, USGS, unpubl. data). This secondary correction was minor for Li, Mg, Mn, Sr, Y, Ba, and Pb with corrections of 0 to 6 %, but more pronounced for U (10 %) and B (24 %). Day–to–day variation of Ba/Ca in MACS–3 relative to the primary standard NIST610 was 5.7 % (2RSD; n = 7) for AG1–2,

AG5, and EG15, and 5.6 % (2RSD, n = 9) for EG3 and EG4, and serves as an indicator for overall data robustness in this study.

### 2.5    Seawater analysis

We used an ICP/MS to measure the major and trace elements in seawater samples at the University of Oxford. To optimize for small sample size, we set up the PerkinElmer NexION 350D instrument in flow-injection mode for analysis. To enable on-line

auto-dilutions and internal standard additions, we connected the instrument to an Elemental Scientific prepFAST M5 autosampler. We doped Indium as an internal standard and used the kinetic energy discrimination (KED) ion guide mode where helium was the cell gas. To monitor for memory accumulation and determine detection limits, we measured all samples in groups of eight and blanks were bracketed in order. Prior to measurement by ICP/MS, we acidified all samples to pH 2 using 12 M HCl. We determined intermediate elements such as Ba using the NexION's flow-injection capability. A significant advantage

for utilising this methodology includes an increase in the TDS tolerance (above the general 2 g L$^{-1}$ ICP/MS threshold), due to the small volumes that are injected. Therefore, the samples were only diluted five times, in order to reduce the TDS concentrations just below 10 g L$^{-1}$. The river water standard SLRS-6 (Canada NRC) was measured periodically throughout the run (n = 4) to determine the precision (3.47 % 2RSD, n = 4) and accuracy (15.04 %) for Ba.

To determine the salinity of each sample, we analysed Na concentration using the instrument's classical setup. We diluted the

samples 100 times using the prepFAST autosampler, in order to reduce the total dissolved solids concentrations to an acceptable range for the instrument (< 2 g L$^{-1}$). To attenuate the signal strength within a range that was suitable for the ICP/MS, we used the NexION's dynamic reaction cell technology, where the rejection parameter "a" was set to a value of 0.016. We measured the sea water standard NASS-7 (Yang et al., 2018) periodically throughout the run (n = 4) and used an average Na seawater concentration taken from Millero et al. (2008) to determine measurement precision (4.12 % 2RSD, n = 4) and accuracy (8.97 %).

We used the conversion between Na concentrations in seawater samples and salinity following Millero et al. (2008).

### 2.6    Oxygen isotope measurements and reconstruction

We extracted oxygen isotope measurements and reconstruction into monthly resolved time series with a precision better than 0.1

‰ from a previous study (Krawczyk et al., 2020) with permission.





### 2.7    Age model

The age model for the oxygen isotope record using powder samples was built using the Sr/Ca ratio as described by Krawczyk et al. (2020). B/Ca showed the best seasonality in all laser ablation ICP/MS profiles in this study and was used for age modelling.

As B/Ca varies inversely with temperature (Dissard et al., 2012), we created age–depth models using MATLAB® by manually assigning the highest and lowest B/Ca to the coldest and hottest month of the year, respectively, based on temperature seasonality extracted from NOAA AVHRR–OISSTv2 High Resolution Dataset with a 0.25° grid size (Reynolds et al., 2007; Banzon et al., 2016). The starting value was assigned as the sampling date of the cores. We applied a linear interpolation between each anchor point followed by another linear interpolation to 12 equidistant points per year to obtain a monthly

resolution. The monthly time series for Ba/Ca is therefore based on the seasonality in B/Ca. Time scale error varies between one and two months because of possible inaccuracies when assigning Sr/Ca and B/Ca values to temperatures as interannual variations can impact the hottest and coldest months in each year. This process allowed us to create a 30 and a 9–year–old $\delta^{18}O_{sw}$ records (AG1–2 and EG3) as well as four 30–year–long (AG1–2, AG5, EG15 and EG4), and a 25–year–long (EG3) Ba/Ca records.


### 2.8    Statistical analyses

We tested variables for normality using the Kolmogorov–Smirnov statistic in MATLAB®. As most key variables were not normally distributed, any test involving comparison to another variable (e.g. correlation or trend), was nonparametric.

All correlations we performed were Spearman's rank correlations when we wanted to assess the presence of a relationship between key variables across a specific period. We performed these on yearly averaged data unless said otherwise, because of very low seasonality within most Ba/Ca time series (Fig. S4). When comparing records with different units (such as Ba/Ca and river discharge), we standardized data by subtracting the mean of the 2006 to 2015 period (common period on all our data) before dividing by their standard deviation (SD) over the same period.  To examine the presence of increasing or decreasing

trends among our records we used the Man Kendall nonparametric trend test indicated by the tau–b statistic. In a second part, we applied Sen's method to estimate the true slope of the linear trend. All trends shown are significant at the 95 % confidence level. When looking at the existence or not of a point in time where our average values increase in a record, we performed a change point analysis (MATLAB, Release 2022b). Although it, assumes normality, in our case the analysis isn't impacted by the non–normality of our variables. Because we are not using the output values as estimators for the parameters of the distribution, but

only the result of the search of the change point (Trauth, 2021) we were able to use this method rather than its nonparametric counterpart. It is important to note that using parametric tests would only reinforce current results.

### 3 Results


### 3.1    Relationship between salinity and reconstructed $\delta^{18}O_{sw}$ with Ba/Ca records

The reconstructed $\delta^{18}O_{sw}$ time series showed similar seasonal to interannual variations over the period of overlap between 2006 and 2015 (Fig. 2a; Krawczyk et al., 2020). A clear seasonality was present, with lower values in autumn/winter (October to January) and higher values in spring (March to May) (Fig. 2a; Fig. S2). AG1–2 $\delta^{18}O_{sw}$ showed overall higher values than EG3

$\delta^{18}O_{sw}$ (-0.28 ‰ compared to -0.38 ‰, p = 0.0031) across their full record length. However, when comparing both records over their common period between 2006 and 2015, we found statistically identical mean $\delta^{18}O_{sw}$ values (-0.4 ‰, p = 0.1459). Both, AG1–2 and EG3 $\delta^{18}O_{sw}$ records were shown to mirror interannual variations in rainfall, sea surface salinity and river discharge (see Tab. 3 in Krawczyk et al., 2020; Fig. S3).

Ba/Ca timeseries showed no clear seasonal cycle across all 5 cores (Figs. 2b to d; Fig. S4). Three of the five Ba/Ca time series show relatively low interannual variations throughout the record (AG1–2, EG3 and EG4), while AG5 and EG15 timeseries do not. Compared to AG5, EG15 exhibits slightly stronger interannual variability (30.5 and 34 % SD, respectively).



When comparing our longest $\delta^{18}O_{sw}$ record (AG1–2) with individual Ba/Ca records, only the EG15 and EG4 records showed a
significant inverse relationship with $\delta^{18}O_{sw}$ across 1984 to 2015 (r = -0.51, p = 0.004 and r = -0.53, p = 0.002). As for Ba/Ca and
SSS, AG5, EG15, and EG4 showed a significant inverse relationship with EN4 salinity across their respective record lengths (r =
-0.49, p = 0.003, r = -0.42, p = 0.02, r = -0.39, p = 0.03, respectively) while AG1–2 and EG3 didn't (p = 0.86 and p = 0.23,
respectively).

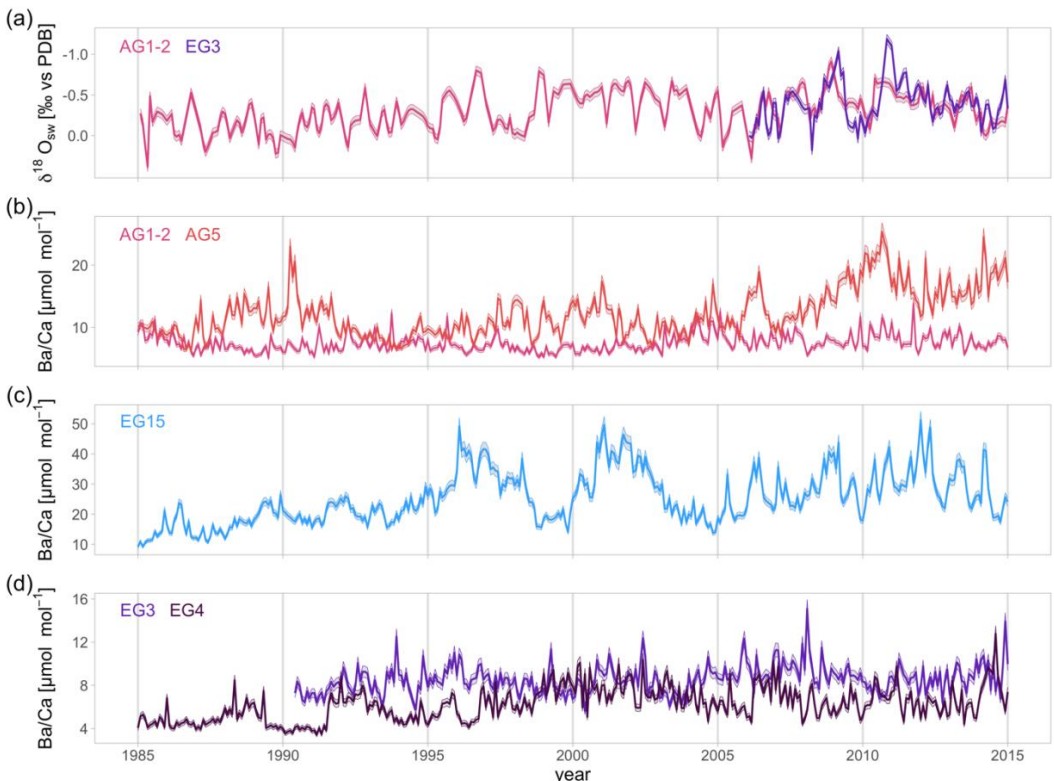


**Figure 2: Monthly interpolated time series of (a) reconstructed $\delta^{18}O_{sw}$ for Anemone's Garden 1 (AG1–2) in pink and Eve's Garden 3
(EG3) in purple (Krawczyk et al., 2020) and Ba/Ca ratios of (b) Anemone's Garden 1 and 5 (AG1–2 and AG5) in pink and red, (c)
Eve's Garden 15 (EG15) in blue and (d) Eve's Garden 3 and 4 (EG3 and EG4) in purple and dark brown, respectively, with shading
indicating analytical error. Note that the y axis on the (a) panel is reversed.**


### 3.2    Ba/Ca trend analysis, composites, and relationship with river discharge

We assessed trends in standardized river discharge and all Ba/Ca records between 1985 to 2015 using Sen's method (Sen, 1968;
Gilbert, 1987; Theil, 1992) (Fig. 3). All six records showed a significant (to the 95 % level) upward trend with slope values of
0.027, 0.067, 0.072, 0.031, and 0.05 for AG1–2, AG5, EG15, EG3 and EG4 Ba/Ca records, respectively and 0.049 for RD. Rates
of change don't seem to have any relationship with distance from the river mouth. AG5 shows the second biggest rate of change
although it is farther away from the river mouth than EG3 and EG4, which both show lower rates of change (0.031 and 0.05,
respectively).



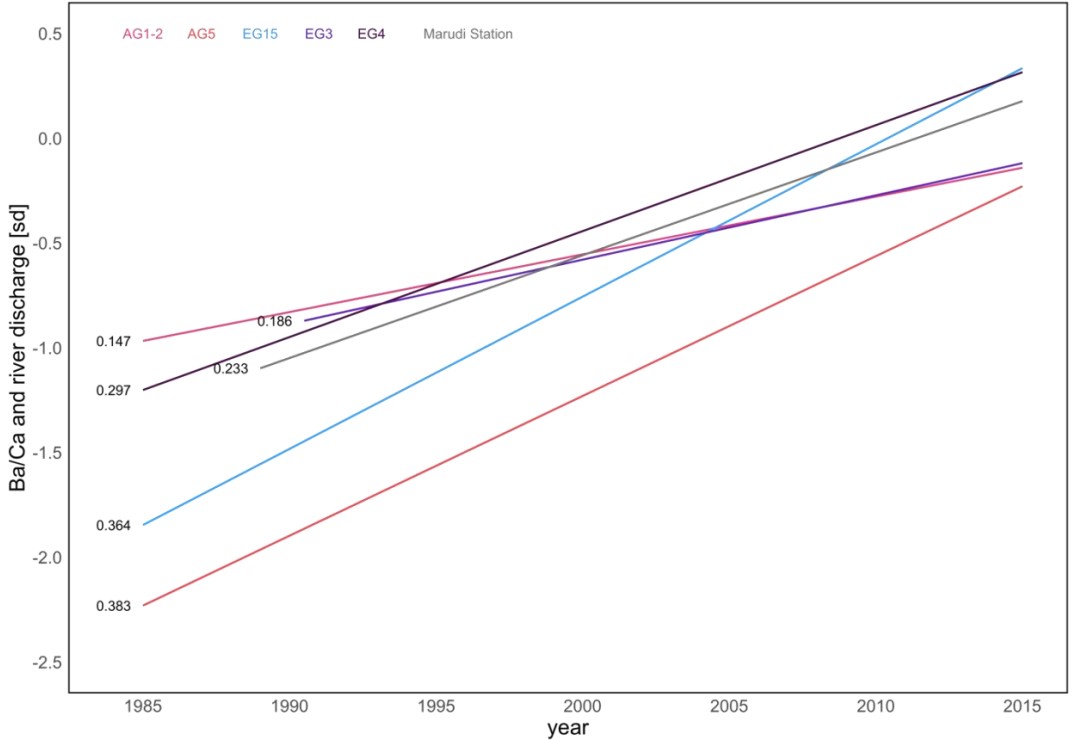


**Figure 3: Trends in the Ba/Ca record of all five cores as well as river discharge using Sen's non–parametric method. The increasing trends are indicated by the Mann–Kendall tau–b statistic next to each record, all results are significant at the 0.05 level.**

To improve the signal–to–noise ratio and eliminate between core variations, we created two monthly Ba/Ca composite records

between 1990 and 2015. The first composite (C1) was created using only records of coral colonies from the closest location to the river mouth (Eve's Garden; EG15, EG3, and EG4). This composite offers the best possible opportunity to trace the river discharge signal and can be compared with the other composite to contrast the signal strength as distance from the river mouth increases. The second composite (C2) includes all five records from both locations at Eve's Garden and Anemone's Garden. It represents an average of what is recorded at the two coral reef sites (see Fig. S4). Here, to avoid a bias toward cores from the EG

colony (three EG cores and two AG cores), we gave all three EG records a weight of 0.1667 and 0.25 for each AG records when we averaged the five. C1 and C2 Ba/Ca composite records show the same order of increase in river discharge (slopes of 0.038, 0.054 and 0.049, respectively). Correlation between composites and river discharge is positive and significant. C1 shows a higher correlation coefficient than C2 (Fig. 4). Unsurprisingly, only C1, being closest to the river mouth, had a significant negative relationship with the $\delta^{18}O_{sw}$ record. When correlated with SSS, both composite Ba/Ca records show a significant negative

relationship, although, this time C2 was stronger.



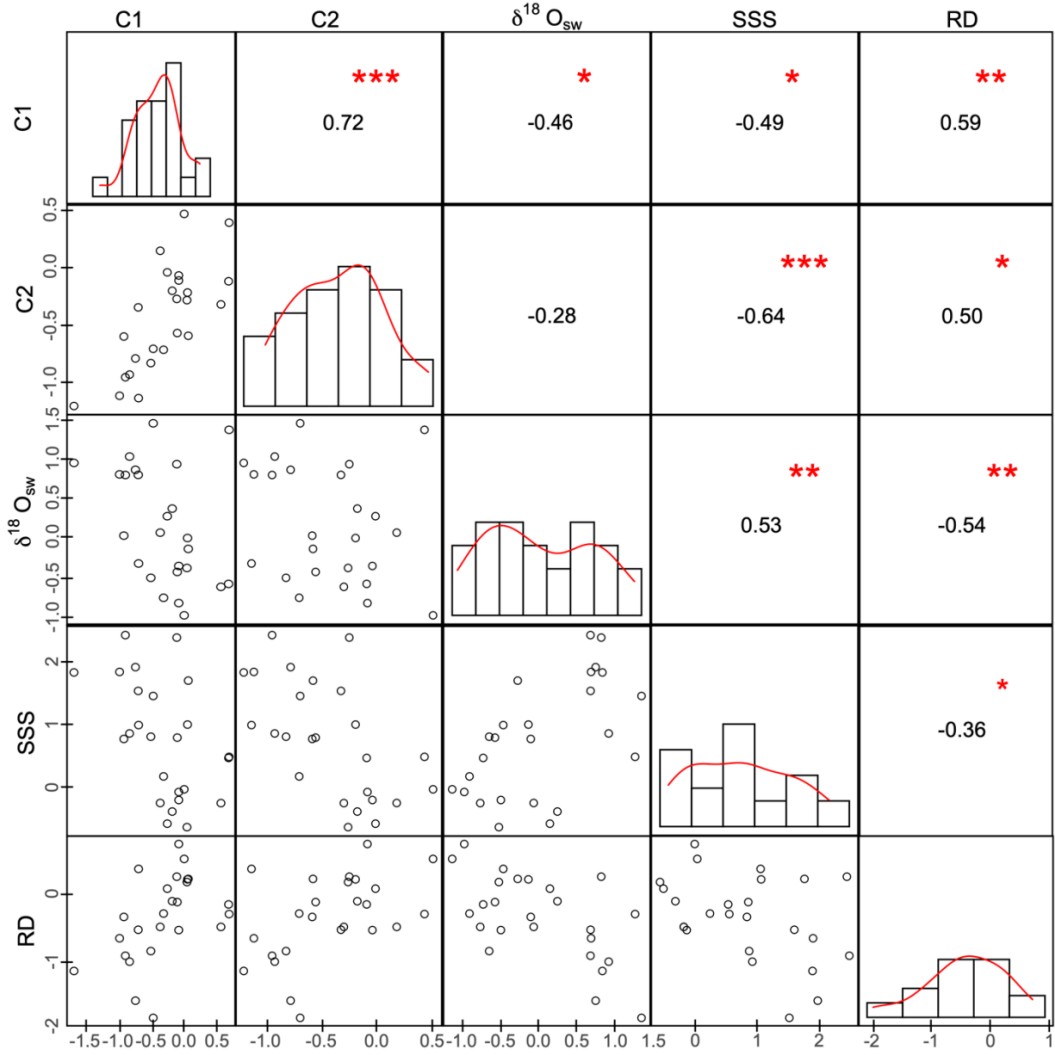

**Figure 4: Correlation matrix between annual standardized coral core–derived proxies for sediment (C1, C2, both Ba/Ca composites) and freshwater ($\delta^{18}O_{sw}$, AG1–2 record) discharge with instrumental data for river discharge (RD) and salinity (SSS) across 1990 to**

**2015. The distribution of each variable is in the centre (as a frequency ranging from 0 to 1), the bivariate scatter plot using standardized units under the diagonal and the value of the correlation as well as significance of the p–value as symbols ("***", "**", "*", ".","") as p–values of 0 and under 0.001, 0.01, 0.05 and 0.1, respectively) over the diagonal.**

When plotted with river discharge, both composites show periods of strong agreement with it, especially C1 (Fig. 5a). When compared with both $\delta^{18}O_{sw}$ records (Fig. 5b), annual to decadal variations are similar, although $\delta^{18}O_{sw}$ records' show slightly larger amplitudes during some periods (e.g. 1996 to 2000 and 2009 to 2011). Interestingly, both $\delta^{18}O_{sw}$ records show an increase from 2010 towards 2015 (indicative of more saline waters) while Ba/Ca values are stable from 2010 onwards.

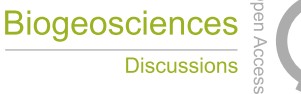



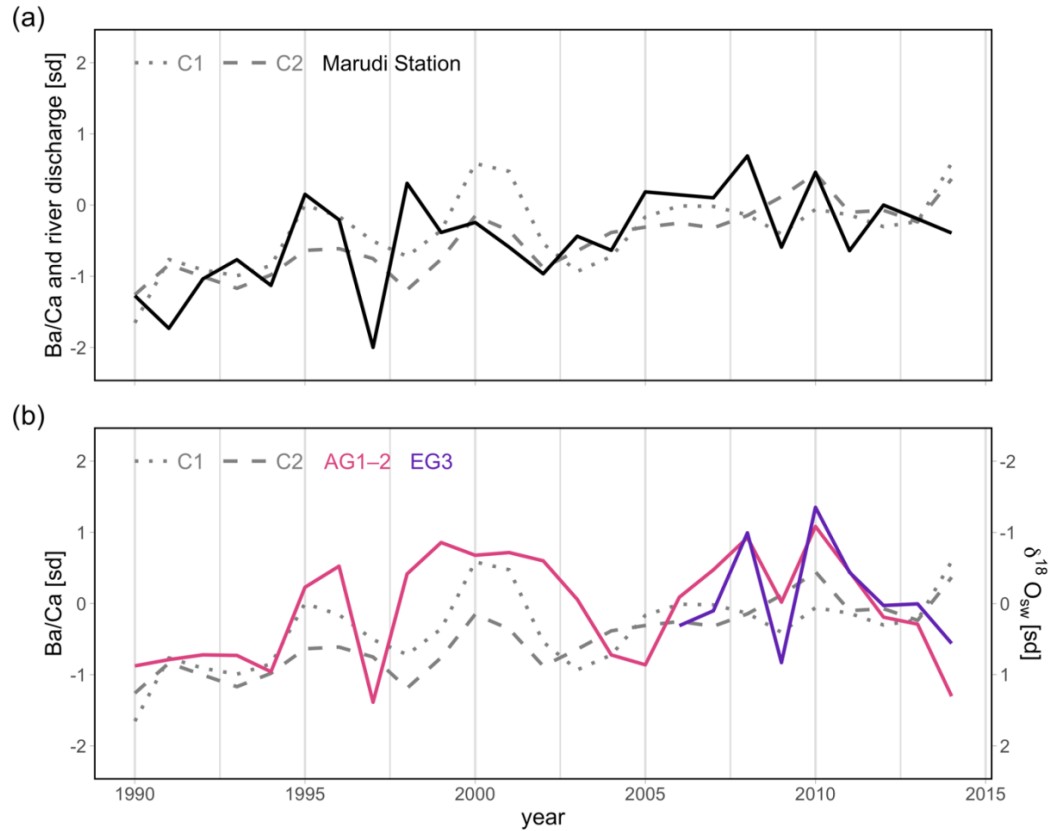

**Figure 5: (a) Mean annual Ba/Ca composite time series C1 and C2 (in grey dotted and dashed lines, respectively) and river discharge using data from Marudi station (in black) and (b) Ba/Ca composites time series with $\delta^{18}O_{sw}$ records AG1–2 (pink) and EG3 (purple), respectively. Note that the $\delta^{18}O_{sw}$ axis is reversed.**

To estimate the point in time separating the time series in two periods, we performed a changepoint analysis based on a significant variation of the arithmetic mean. River discharge showed a changepoint in 1995 (Fig. 6a). When subjected to the same analysis, C1 and C2 had different change points in 1995 and 2004, respectively (Fig. 6b and 6c). When performed on both EN4 SSS and the AG's $\delta^{18}O_{sw}$ record, the analysis showed changepoints in 2007 and end of 1998, respectively (Fig. S5).



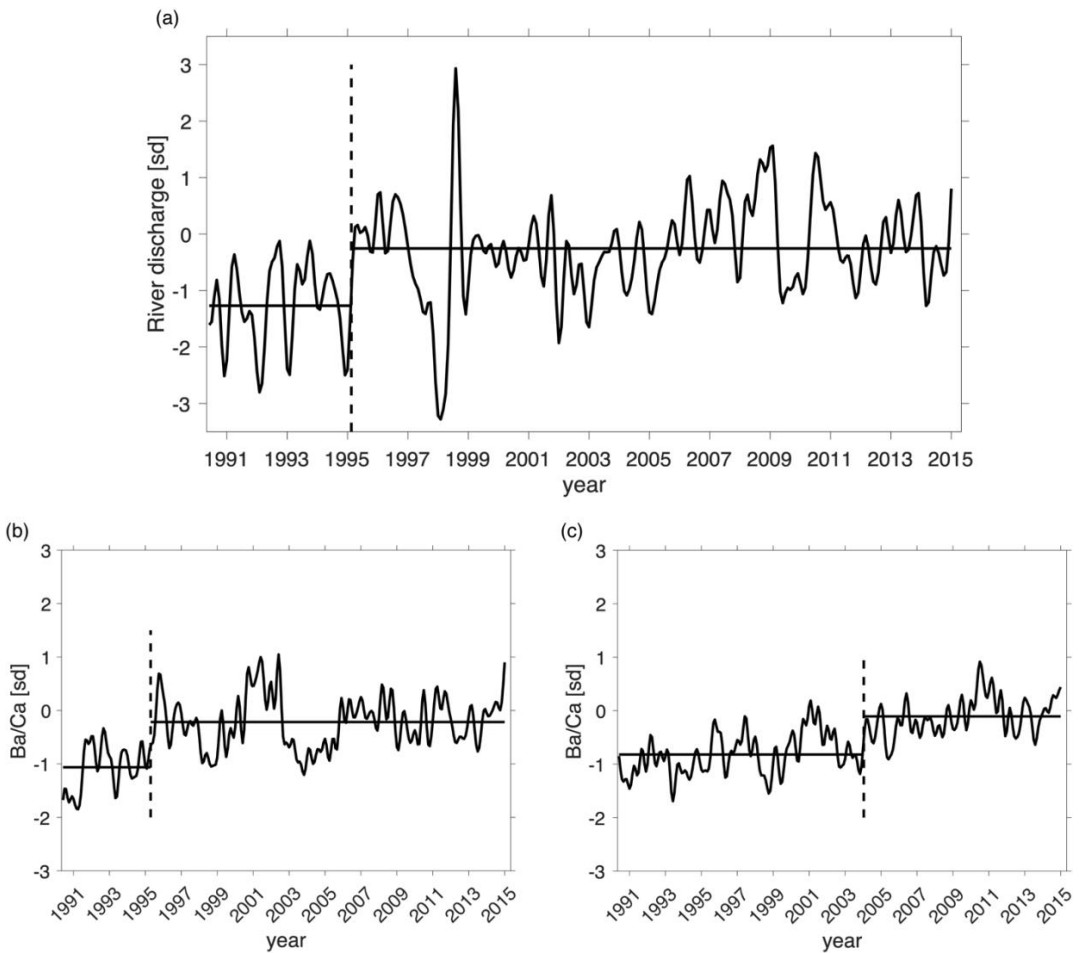

**Figure 6: Change point analysis (MATLAB, Release 2022b) based on significant arithmetic mean change for (a) Marudi station's discharge as well as both composite records, (b) C1 and (c) C2.**


### 3.3 Relationship between Ba/Ca, ENSO and deforestation

Our composite Ba/Ca monthly records show significant negative correlation with the Niño3.4 index (r = -0.31, p < 0.001, N = 296) with C1 and with C2 (r = -0.19, p = 0.001, N = 296), albeit weak. The relationship between Niño3.4 and both composites are weaker than Niño3.4's relationship with river discharge (r = -0.38, p < 0.001, N = 313) or salinity, although the C1 and C2 response appears to be higher after a prolonged period of very intense El Niño's in 1998, 2005 and 2010 (Fig. 5).

Satellite–derived spatial deforestation data were mapped to show the extent of deforestation through the available time record in the Baram River catchment and were converted into an annual deforestation time series starting from 2001 to 2019 (Fig. 1a). As the data are annually resolved and did not include data prior to 2000, we couldn't standardize them the way all other times series were. However, this record can still be used to determine periods of relatively faster deforestation than usual in the Baram River catchment during this time span. Annual deforestation rates increased dramatically for ten years from 2002 to 2012 by 534 %, with a 43 % drop in 2010 compared to the previous year, increasing again in 2012, before stabilising at a baseline an order of magnitude higher than pre–2009. Post–2009, the increased deforestation rates were largely located in the low altitude areas close to the coast (Fig. 1a) and were probably related to the one million ha of palm oil plantations by 2010 state target established in 2007 (Tsuyuki et al., 2011). Since 2012 to 2019, annual deforestation rates across the Baram catchment show a decreasing trend (39 %) except for a small increase in 2014 and 2018. The significant decrease in annual deforestation rates in 2010 compared to





the previous and following years did not match with a particular ENSO state. It occurred during an El Niño state until March 2010 before becoming a La Niña state for the rest of the year. Furthermore, both rainfall and river discharge did not show any significant difference between 2009 and 2010 (p = 0.436 and p = 0.583, respectively), nor between 2010 and 2011 (p = 0.795 and

p = 0.312, respectively).

Coral core Ba/Ca records and actual rates of deforestation were only partially aligned. When we compared both Ba/Ca composite records between 2009 and 2010 with deforestation, only C2 showed a significant increase (p = 0.006) while C1 had no significant change (p = 1) during the same period. Conversely, in 2011, when yearly deforestation increased again from 12674 to 34048 Ha, only C1 showed a significant increase between 2010 and 2011 (p = 0.01) while C2 didn't (p = 0.312).


## 4   Discussion

### 4.1     Fidelity of Ba/Ca ratio as a record of sediment discharge

Given the proximity of coral colonies from both reefs relative to the outflow direction of the Baram River, we anticipated this to

be the main input of sediment in river discharge, and therefore the main driver of Ba/Ca in coral skeletons. However, when we correlated Ba/Ca data against river discharge, only two individual records showed a significant positive correlation (EG1 and EG3) at a 95 % confidence level.

Furthermore, combining records to create two composites greatly improved the ability for coral to record sediment in river discharge as both C1 and C2 showed greater correlations than any individual core with river discharge (0.59 and 0.5,

respectively). Given the magnitude of the improvement, high frequency, monthly, and intercolonial variations seem to play an important role in coral record variations that should not be overlooked when using Ba/Ca as a proxy. Indeed, small seasonal and monthly variations of river discharge within a year rely on a perfectly built age model to be reflected accurately throughout the coral record. Additionally, constant hydrodynamic conditions would be necessary for a constant amount of river water and sediment to consistently reach coral colonies regardless of season. This is far from reality as river freshwater and sediment

transport to nearshore waters in this region is strongly influenced by local monsoon–induced winds (reversal onset of each monsoon period) and their impact on surface currents in the South China Sea and the Malaysian coast (Krawczyk et al., 2020).

We expected differences between records from Eve's Garden and Anemone's Garden to be indicative of the distance between each location and the river mouth (i.e., lower absolute values in Ba/Ca in coral colonies furthest away from the river mouth

compared to closer colonies) due to dilution. However, our seawater samples collected in October 2019 between the Miri River mouth and coral reef sites AG and EG revealed Ba/Ca$_{coral}$ to Ba/Ca$_{sw}$ ratios higher (2.2 and 3, respectively) than previous studies (Lea et al., 1989; Lea and Spero, 1992) and no significant difference in seawater Ba/Ca concentrations (Fig. S6). On average, skeletal Ba/Ca values were higher in cores from the EG location than Ba/Ca values in cores from AG (13.4 vs 9.9 µmol mol$^{-1}$), however this was due to comparatively high values in one of the cores, EG15. When EG15 was removed from the analysis,

Ba/Ca values from EG were, on average, lower than cores from AG (7.4 vs 9.9 µmol mol$^{-1}$). This difference, opposite to what was expected, could be explained by inter–colonial variability in growth rates and/or Ba uptake (Tanzil et al., 2019) due to local–scale hydrodynamic processes influencing sediment delivery. These explanations were also put forth to explain similar differences in two studies that showed highly variable Ba/Ca values between reef sites and withing reefs at various distances from the river mouth in the Great Barrier Reef (Lewis et al., 2007, 2012). Indeed, a significant number of previous studies have

shown coral–based geochemical records such as Ba/Ca to be influenced by other factors depending on the location. These include local to regional scale processes influencing oceanography, as well as hydrodynamic conditions such as complex geography (islands), local currents, and upwellings (Lea et al., 1989; Montaggioni et al., 2006; Lewis et al., 2012), local runoff/groundwater discharge from coastal streams (Swarzenski et al., 2001; Gonneea et al., 2011; Jiang et al., 2018), exogenous biotic processes such as phytoplankton blooms (Lewis et al., 2012; Jiang et al., 2018; Tanzil et al., 2019), or endogenous biotic

processes like coral spawning (Sinclair, 2005; Gagan et al., 1996). Additionally, several of these processes can lead to sediment resuspension capable of further altering the Ba/Ca record (Hanor and Chan, 1977; Colbert and McManus, 2005; Grove et al., 2012; Tanzil et al., 2019). However, when considering our study site, multiple explanations can be discarded as the location does not feature small islands and isn't impacted by upwellings as they were shown to be present only on the north–western coast of Borneo during the beginning of winter (Yan et al., 2015). Additionally, the lack of clear cyclicity in Ba/Ca peaks does not favour





phytoplankton bloom or coral spawning impacts, which do not appear to have had a strong enough impact on coral biochemistry to appear in the record (Sinclair, 2005), therefore leaving only small–scale local currents, local runoff, and groundwater discharge as possible explanations for that discrepancy. Nevertheless, mean annual Ba/Ca composite records tracked river discharge and our seawater transect data do not rule out the possibility that seawater Ba/Ca ratios and therefore skeletal Ba/Ca can be in a similar range.


### 4.2 River discharge as a driver of sediment runoff and Ba/Ca

All Ba/Ca records showed a similar upwards trend and rate of change to that of river discharge from the nearby Baram River (Fig. 3). As shown in Fig. 6, the increase is comparable between C1, C2, and river discharge, however the timing of the change point is different between river discharge and C2. Adding AG cores into the composite shifted the change point timing further

towards the present and the nine–year lag between the two was too large to be explained solely by the distance between AG and the mouth of the Baram River. We argue that a yet unquantified "threshold effect" may prevent the riverine Ba signal from reaching colonies furthest away in amounts proportional to the distance between the river mouth and individual colonies. This would imply that for AG colonies to record a significant increase in Ba/Ca, a signal greater than the one required for EG colonies is required, even when taking distance into account, thus explaining the difference in change points.

As suggested by several previous studies in tropical watersheds, coral Ba/Ca records are strongly linked to watershed hydrology, climate variability, forest cover, as well as population growth (Maina et al., 2012) and seem to respond to a common signal (Grove et al., 2012). ENSO was shown to be an important driver of regional hydroclimate in Sarawak and across Borneo (Tangang and Juneng, 2004; Juneng and Tangang, 2005; Sa'adi et al., 2017b). Previous analysis of coral–derived $\delta^{18}O_{sw}$ from Miri also revealed a strong influence of ENSO on river discharge and salinity (Krawczyk et al., 2020). Our composite Ba/Ca

monthly records show significant negative correlation with the Niño3.4 index, albeit weaker than their relationship with river discharge or salinity (Krawczyk et al., 2020). Enhanced soil erosion of accumulated sediments during dry conditions following intense El Niño drought periods have been reported by skeletal Ba/Ca studies in the Great Barrier Reef (McCulloch et al., 2003; D'Olivo and McCulloch, 2022). C1 and C2 showed elevated levels of skeletal Ba/Ca following very intense El Niño years, *e.g.* between 1997/98 to 1999–2001, after 2003–2005, and after 2010. Other local factors, such as catchment land use change, may

have masked the ENSO relationship with sediment runoff in recent decades.
Nevertheless, results here show a strong significant relationship between coral Ba/Ca and river discharge, and although the change point timing is different, the comparable increase between both coral Ba/Ca composites and river discharge is very interesting as it could possibly rule out an increase in Ba content in riverine waters in favour of a river discharge increase accurately recorded by corals on a multi decadal time scale.


### 4.3 Linking Ba/Ca with deforestation

Between 1973 and 2015, 2.39 of the 9.22 Mha of forested area was logged, out of a total land area of 12.4 Mha in the Sarawak region, representing an almost 26 % decrease in total forest area. There are clear interregional differences in forest loss within Borneo, as the entire island saw a 33.4 % decrease in forested area in that same 45 year period (Gaveau et al., 2016), highlighting

interregional differences within Borneo, although our data seem to show a Sarawak's deforestation rates plateau from 2013 to 2019.
When subjected to Sen's trend analysis, precipitation, the main driver of river discharge, did not show any significant trend across the record (Fig. S7), suggesting that the river discharge increase was not due to local hydrological changes but rather

related to freshwater storage or runoff change. In different forested catchments in East Africa, trend analyses and modelling studies have shown river discharge and runoff increases to be linked with forest cover loss (Guzha et al., 2018). Similarly, long term river discharge increase with no significant evidence of rainfall increase was attributed to land use changes in South America's Parana River catchment (Lee et al., 2018).
Several studies demonstrated the impacts that land use and deforestation can have on land erosion and river flow rates, such as

increased terrigenous sediment concentration in rivers, lakes, and ultimately mixing zones, eutrophication, bioerosion and smothering of local aquatic life, once river waters reach the marine environment (Neil et al., 2002; Shi et al., 2013; Restrepo et al., 2015; Karamage et al., 2016). Furthermore, recent studies have focused on the difference in freshwater discharge in deforested and forested catchments and have shown that forest loss is accompanied by increased sediment runoff, river





discharge, and shifts in the months of peak discharge events while also stressing the non–linearity of freshwater discharge response to deforestation (Sajikumar and Remya, 2015; Guzha et al., 2018; Lee et al., 2018; Booij et al., 2019). Although none of the aforementioned studies focus on Malaysian Borneo, some are applicable to this region and similar processes have been observed in the Sarawak region (Vijith et al., 2018a). Furthermore, a recent study in our sites showed comparable consequences (high bioerosion and algal cover) for the marine environment from sediments in river discharge (Browne et al., 2019).

Results indicate that continuous increase in deforestation for the past four and a half decades is most likely responsible for the river discharge increase recorded by Marudi station data and the corals Ba/Ca signals. The Ba/Ca patterns have been impacted by enhanced soil erosion that affected water and sediment storage in the Baram catchment and in the whole of Sarawak (Vijith et al., 2018a), as observed in previous studies utilizing coral geochemistry (Prouty et al., 2008; Yu et al., 2015).


## 5 Conclusion

We examined the use of coral skeleton Ba/Ca ratios as a proxy for sediment in river discharge (terrestrial runoff) off the coast of Malaysian Borneo using multiple coral colonies from two reefs within the Miri–Sibuti Coral Reef National Park, with samples obtained at different distances from the river mouth. We assessed correlations with river discharge and rainfall, overall trends and interannual to decadal variability. We found that several coral Ba/Ca records (the more the better) were necessary to reconstruct the sediment in river runoff history as inter–colony differences were strong, especially at reefs further away from the
river mouth (>10 km). Performing changepoint analyses further clarified these and outlined the importance of record replication when using this proxy in such coastal mixing zones.

Nonetheless, Ba/Ca ratios of all colonies studied here showed similar increases to that of river discharge on yearly to decadal time scales, as well as significant correlations once we built composite records. These findings are best explained by increased land use through deforestation leading to increasing amounts of freshwater and sediment runoff into the coastal system. Our
analysis stresses the importance of several processes that need to be taken into account when considering using coral Ba/Ca ratios as a proxy for sediment discharge including: (1) distance from the river mouth, (2) record replication, (3) surface ocean current seasonal variability, (4) hydrodynamic behaviour of the river plume, (5) trace element behaviour in the mixing zone, and (6) availability of instrumental records of river water data.

We recommend future studies of Ba/Ca$_{coral}$ ratios should be coupled with trace element seawater and river water sampling on a transect between the coral locations and the river mouth to assess trace element behaviour along the ridge to reef path. In addition, Ba/Ca ratios in corals and surrounding waters should be compared across seasons when possible. Furthermore, using δ$^{18}$O$_{sw}$ composite records from the same colonies to track river discharge could be beneficial to disentangle the influence both river discharge and sediment in river discharge have on Ba/Ca$_{coral}$ ratios by recording information about local freshwater
discharge. Although the purpose of this study was to test the validity of the Ba/Ca proxy in this region during a period where instrumental records are available before applying it further back in time, we stress the necessity of having accurate instrumental records of river discharge (and sediment load if possible) as close to the river mouth as possible to compare to Ba/Ca records. This is crucial as river discharge may not represent the main source of Ba in the coastal system in some locations.

Larger scale studies are needed to expand coral geochemical reconstructions both spatially and temporally to include the
beginning of the 20$^{th}$ century as well as another location in the national park so that we can better elucidate the pathway of Ba with distance from the river mouth and further study the effect human development has had on Malaysian Borneo.

## 6 Code availability

The code used to interpret global forest change data can be found in the supplementary information section of the paper.



## 7 Data availability

The coral proxy data from this publication will be archived after publication with the public NOAA WDC data portal at https://www.ncdc.noaa.gov/data-access/paleoclimatology-data/datasets.
The paper uses data from the global forest change dataset available at: https://storage.googleapis.com/earthenginepartners-hansen/GFC-2020-v1.8/download.html.

## 8 Author contribution

WN analysed the data and wrote the manuscript.
JZ, AB, RN, NB, NE, JM, PH and KR contributed to manuscript review.
MP extracted, analysed, and plotted satellite data.
KR, BM, NB, PH and NE collected and processed the samples.


## 9 Competing interests

The authors declare that they have no conflict of interest.

## 10 Acknowledgments

We would like to thank the Co.Co. Dive Miri team, Toloy Keripin Munsang, Valentino Jempo from the Sarawak Forestry Department as well as Daisy Christy Saban who made fieldwork possible.
Fieldwork was approved by the Sarawak Forestry Commission (permit no (61)/JHS/NCCD/600-7/2/107). Coral cores were imported under CITES licence number 002259.

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
