# Peer review of "Massive corals record deforestation in Malaysian Borneo through sediments in river discharge"

_Biogeosciences, 2022_

## Author Response (AR1)

**Reviewer 1:**

**(i) Ba/Ca: I was wondering if you could provide some more information on Ba/Ca.**

**- What are the possible sources of Ba/Ca?**

Possible sources of Ba other than river discharge include algal blooms and productivity, coastal upwelling, and submarine groundwater discharge (cited lines 432 to 437). However, the lack of any case studies mentioning Ba addition to the coastal seawater environment other than river sediment discharge made those sources unlikely or presently unknown. Furthermore, a previous study cited in the discussion (line 439) established that the only upwelling region in Borneo is further North, while the lack of studies showing any existing submarine groundwater discharge as well as the distance separating our sampling sites and the coast (7 and 11 km) have led us to discard it as a potential source of Ba.

**- What is the ambient value of Ba/Ca in seawater? Does this change over time for any reason?**

We collected surface seawater samples on October $11^{th}$ 2019, so unfortunately (due to Covid-19 travel restrictions), for now, we only have values for one transect from the Miri River to the coral reef locations (see Supplement Fig. S6), from the monsoon transition season (end of dry season/start of wet season). Ba/Ca seawater values were 4.41 µmol/mol (± 0.08, N = 4) across duplicates for both coral reef sites. Values of inshore seawater Ba/Ca (2-4 km from river mouth) were 17.74 ± 0.69 µmol/mol (N = 2) and river mouth (200 m upriver) samples were 59.82 ± 1.29 µmol/mol (N = 3). Our seawater Ba/Ca data therefore support conservative mixing of Ba/Ca with distance from the Miri River. Based on the premise that corals incorporate trace elements into their skeletons in concentrations proportional to these found in seawater, and the fact that coral skeletons show a Ba/Ca increase, we would assume that Ba/Ca$_{seawater}$ baselines in 1985 would be lower than present values. On a seasonal timescale, we would expect the ambient value in the seawater to slightly change depending on the monsoon season as wind currents reverse from southeaster to north-western which could influence the amount of river water (and sediment) reaching both sites located south of the river mouth. We currently have a seawater monitoring program in place for both Baram and Miri rivers that will ensure sampling in dry and wet season between 2022 and 2023.

**- What is the geology of the catchment area? Is it fairly uniform or could deforestation in different catchment areas result in different fluxes of Ba/Ca in the river?**

This catchment area of the Baram River mainly consists of meta-sedimentary to sedimentary rocks such as shale, sandstone, and minor amounts of limestone (aged from Paleogene to Miocene) as mentioned in the introduction. The upstream of the Baram River mainly consists of Oligocene to Eocene meta-sedimentary and sedimentary rocks while the downstream region mainly consists of river and coastal alluvium of Quaternary age. The Alluvium sediments include the peat deposits common in the northern part of Sarawak (Anderson, 1964). In addition, a relatively smaller portion in the catchment area is covered by volcanic rocks in the southwest part of the Baram River basin. As for the uniformity, Vijith et al., (2018) show a very uniform soil erodibility factor across their study site, which is located within the catchment. The study site, however, only represent a part of the catchment, for the rest of it, we can only assume the erodibility factor to be relatively similar.

Nevertheless, it would be reasonable to assume that some very localized areas of extremely high erodibility would lead to higher fluxes, depending again on the geology.

**- Are there any built structures on the river path that could track sediments? Has this changed in the past? Could this impact Ba/Ca-sediment-discharge volume relationship?**

We assume that the reviewer meant trapping sediments instead of tracking. As far as we are aware, there is no built structure allowing sediment trapping. There was initially a project of a dam on the Baram River, 250 km inland of Miri, but it was scrapped because of public outcry as it would flood ancestral lands. As far as we know, there isn't any structure that could lead to some sediment retention or the opposite in this catchment. A dam would certainly impact the relationship between Ba and discharge volume, although we fail to see how it could impact the Ba to sediment discharge volume relationship as there would be very few ways for such a structure to selectively impact Ba more than other elements to our knowledge.

**- Are there other coral measurements from the region that are not downstream of this river, do they show similar upward slopes in Ba/Ca or other proxies?**

Unfortunately, we do not have any coral proxy records in the near region that is not (at least partially) influenced by the Baram River, although seeing if they show a similar trend would be interesting.

**(ii) Coral age model: I'm a little confused for how the seasonal age model was developed and I was hoping you could provide some more details**

**- Did the coral grow with the same rate across all seasons?**

Like most corals from the genus *Porites* the growth rate is usually found to be somewhat affected by light and sediment availability making them slightly biased towards summer, although comparably to most studies involving *Porites* records, for our monthly age model we assumed the growth rate to be linear.

**- How are B/Ca and temperature related?**

B/Ca is believed to be inversely correlated to seawater temperature as shown by (Fallon et al., 2003), although we acknowledge that it is not the best or most used proxy for seawater temperature. B/Ca was used instead of the most traditionally used Sr/Ca as the Sr results from laser ablation ICP-MS weren't reliable enough.

**- So the previously published Sr/Ca based age model provided only an annual age model?**

The previously built age model was monthly as it was based on wet chemistry ICP-MS for which Sr/Ca worked very well, see (Krawczyk et al., 2020). We used the published monthly Sr/Ca and oxygen isotopes data from the previously publication by Krawczyk et al. (2020) as stated in the methods section.

**-And just out of curiosity, which MATLAB script was used for developing the seasonal age model? Why not use QAnalyseries?**

We used a script that we wrote using the function "interp1" to do the linear interpolation based on the chosen anchor points. We will make the code available publicly upon publication.

The main reason behind our use of a "homemade" MATLAB script rather than QAnalyseries was the presence of regular bugs within the software reported by some co-authors, further, using a script we wrote allowed us to be more in control of what happens behind the scenes. We did compare the first output from our script to Analyseries and obtained the same result, which comforted us to keep using the script for the other age models.

**(iii) Changepoint analysis:**

**- Was the code told how many changepoints to look for in the dataset? Was this set to 1 for all the analyses?**

Yes, in addition to being able to select the metric for which we wanted to find a changepoint (e.g. average, standard deviation, etc), one input of the function is the number of changepoints to find in a given timeseries. We chose one (the most significant) for all analyses.

**- Just as a check to make sure that the changepoint isn't resulting from coral growth, have you tried doing the analysis on multiple proxy measurements from the same coral core. They could all still be changing in response to an external climatic condition but it would provide more information.**

We only considered Ba/Ca in this study and only applied it to Ba/Ca records in our main analysis, besides the $\delta^{18}O_{sw}$ for the E core that showed a different changepoint in the SI. However, we did test that theory during preliminary analysis and established that both Sr/Ca and B/Ca timeseries did show different changepoint when compared to their Ba/Ca counterpart.

**(iv) What is the relationship between deforestation and sediment erosion in this region? Presumably increased deforestation causes increased sediments in the river, but is this linear? Is this different in the different catchment areas? Particularly since all the deforestation doesn't result in barren land but rather in replantation. Borneo also has peatland and sometimes it is an additional case of draining peatlands for plantation that presents a different situation than other locations around the world. Is conservation active in the region? When did conservation efforts start? Would you expect to see this reflected in the Ba/Ca data?**

Great questions, yes, a previous study cited in the introduction (Vijith et al., 2018) did show how several heavy logging areas created erosion hotspots although this study, specifically, did not investigate the amount of sediment reaching the rivers. As you said, the relationship is believed to be nonlinear (based on other studies in different catchments where deforestation was linked to a nonlinear river discharge increase cited in section 4.3). Deforestation in different catchments will certainly lead to stronger or weaker increase in river discharge depending on the rate, geology and presence or absence of replantation. But, until now, these studies and our results seem to convey that heavy deforestation leads to an increase. Its intensity is probably heavily dependent on the parameters cited above.

As you mentioned, in this specific case, most deforestation is succeeded by palm oil plantations, which does dampen its effect although after some time needed to reach stable condition, while conversely, blackwater rivers from peatland draining could lead to disproportionate effects in the other direction. Although in Sarawak's case, most peatlands consist of more than 90% organic matter, making them a less probable source of trace metals such as Ba.

As for conservation efforts, although research papers focusing on the necessity of conservation efforts in Borneo are not rare, very little is being done on a jurisdiction level. Some foundations such as Borneo Nature Foundation focusing on protecting the rainforest left exist although the scale of their action is still too small to see a discernible impact and might be overshadowed by vast constructions projects such as the Pan Borneo Highway.

Whilst dates of the commencement of forest conservation in Malaysian Borneo are vague, 31% of the 40 Mha of forest in Borneo are within designated Protected Areas (PA) (World Wide Fund, 2017), with 8 – 10 % of these PAs residing within Malaysian Borneo (Primack, 1991; Meijaard and Sheil, 2007). By 2015, Sarawak had lost 25 % of the 9 Mha of intact forest in 1973 (Gaveau et al., 2016) with deforestation within PA occurring but the extent of which has not been quantified (Gaveau et al., 2014).

**Typing errors etc.**

Line 25: What in the Porites recorded a very similar upward trend? Changed "*Porites* sp. records" to "Ba/Ca$_{coral}$ records" for clarity.

Line 40: Just a pet peeve :) There is no such thing as 'satisfying a global demand', best to leave it at '... since the middle of the 20th century for oil pal and pulp wood'. We assume the reviewer meant it is very difficult to satisfy global demand, as there is a worldwide demand for oil palm and pulpwood because of the multitude of products they are used in as well as the scale of these products. Changed "since the middle of the 20th century to satisfy the global demand" to "since the middle of the 20th century driven by global demand".

Figure 1: can you add some figures to show deforestation versus river discharge versus rainfall amount. Annual data, and then monthly over an annual cycle if this is available for rainfall and river discharge. 1a, the colour of the river versus forest is not clear in the print. 1c, the red shaded box demarcating the region of interest is not clear in the print. We could show the annual data although the common period to all three records would only be across 2001 to 2014. We don't have access to monthly deforestation data but could show rainfall and river discharge across an annual cycle (see attached file). Both could be in the supplements.

[Figure]

[Figure]

Increased contrast between forest and river colours. Reduced the transparency of the red shaded box for easier visibility.

Line 135: '..... have been exposed to increased sediment loads....' Changed "has" to "have".

Line 245: Expand acronym TDS here. This has instead been done later in the next paragraph. Changed "an increase in the TDS tolerance" to "an increase in the total dissolved solids (TDS) tolerance".

Line 285: Remove comma... 'Although it assumes normality,...' Comma removed.

Line 300: Specify that the d18O time series of the 5 cores collected for this study, it takes a second to realise that you are talking about the 5 cores. There seems to be a misunderstanding here, there are only two $\delta^{18}O_{sw}$ records as mentioned in the first paragraph in 3.1. Added "two" in line 299 for easier understanding.

Figure 2b and 2d: The contrast in colours between the two curves in the respective figures is not clear in the printouts. Increase contrast or use dash-dot lines etc. Changed the colour palette across the paper for a better contrast between each colour.

Line 395: Toward the end of this paragraph, the information is suddenly given in number of hectares rather than a percentage change as given elsewhere. Maybe best to give this in percentage as well. Changed "when yearly deforestation increased again from 12674 to 34048 Ha" to "when yearly deforestation increased again by 169 %".

Line 425: typo correction '...within reefs at various distances...' Changed "withing" to "within".

Add page numbers to the supplementary information file. Added the page numbers.

**References**

Anderson, J. A. R.: The structure and development of the peat swamps of Sarawak and Brunei, J. Trop. Geogr., 18, 1964.

Fallon, S. J., McCulloch, M. T., and Alibert, C.: Examining water temperature proxies in Porites corals from the Great Barrier Reef: a cross-shelf comparison, Coral Reefs, 22, 389–404, 2003.

Gaveau, D. L. A., Sloan, S., Molidena, E., Yaen, H., Sheil, D., Abram, N. K., Ancrenaz, M., Nasi, R., Quinones, M., Wielaard, N., and Meijaard, E.: Four Decades of Forest Persistence, Clearance and Logging on Borneo, PLoS ONE, 9, e101654, https://doi.org/10.1371/journal.pone.0101654, 2014.

Gaveau, D. L. A., Sheil, D., Husnayaen, Salim, M. A., Arjasakusuma, S., Ancrenaz, M., Pacheco, P., and Meijaard, E.: Rapid conversions and avoided deforestation: examining four decades of industrial plantation expansion in Borneo, Sci. Rep., 6, 32017, https://doi.org/10.1038/srep32017, 2016.

Krawczyk, H., Zinke, J., Browne, N., Struck, U., McIlwain, J., O'Leary, M., and Garbe-Schönberg, D.: Corals reveal ENSO-driven synchrony of climate impacts on both terrestrial and marine ecosystems in northern Borneo, Sci. Rep., 10, 3678, https://doi.org/10.1038/s41598-020-60525-1, 2020.

Meijaard, E. and Sheil, D.: A logged forest in Borneo is better than none at all, Nature, 446, 974–974, 2007.

Primack, R. B.: Logging, Conservation, and Native Rights in Sarawak Forests, Conserv. Biol., 5, 126–130, 1991.

Vijith, H., Seling, L. W., and Dodge-Wan, D.: Estimation of soil loss and identification of erosion risk zones in a forested region in Sarawak, Malaysia, Northern Borneo, Environ. Dev. Sustain., 20, 1365–1384, 2018.

World Wide Fund: The Environmental Status of Borneo, Heart Borneo Programme, 2017.

**Reviewer 2:**

Line 26: "significant positive correlations" Is this a monthly correlation? Annual? Inter-annual? Added "annual".

Figure 1a: On my copy, the blue color (Baram basin rivers) and the green color (Forest) are very similar. Can they be modified to be more distinct from one another? Increased the contrast between the two colours.

Figure 1 caption: I think there should be a citation at the end of the caption for the raster data. Added citation.

Lines 154-156: It would be useful to know the standard deviation of the range in seasonal temperatures and rainfall. The manuscript lists a "2.5°C" range between warm and cool months. 2.5 ± 5 is different from 2.5 ± 1. Changed from a difference between average monthly values to an iterative difference between monthly maxima and minima for each year. Changed to "3.1 ± 0.7 °C" and "405 ± 125 mm".

Line 159: Keep the discharge units as similar as possible between the low discharge and high discharge months. For example, just change the high discharge values to 2550, 3130, and 3080 so they are more easily comparable with the low discharge value of 2072. Good point, changed to "2250, 3130 and 3080".

Line 176: datasets could use a citation here. Added the necessary citation "(Sa'adi et al., 2017a)".

Line 263: I think some words are getting switched around here to make this sentence unclear. Changed "The age model for the oxygen isotope record using powder samples was built using the Sr/Ca ratio as described by Krawczyk et al. (2020)" to "The age models of the $\delta^{18}O$ records were built using the Sr/Ca seasonality as described by Krawczyk et al. (2020)".

Line 438: "upwellings" should be "upwelling". Changed "isn't impacted by upwellings as" to "isn't impacted by any upwelling as".

Line 467: "between both" could be "among". Changed "comparable increase between both coral Ba/Ca composites" to "comparable increase among coral Ba/Ca composites"

**Editor:**

1. **Here, I would especially aim to improve the description of:**

**Possibly sources and variability in Ba/Ca more clear (although this was discussed partly in the manuscript, potential impacts of sediment trapping and variability in soil Ba run-off because of geological variability could be elaborated on).**

In our response to the reviewers, we have slightly expanded on the possible sources of Ba into the coastal system and they are now listed in lines 438 to 445.

Added "Additionally, some processes or terrain properties further contribute to the non-linearity of the relationship: these include sediment trapping and geological variability of the catchment, which leads to disproportionate Ba concentrations in discharge waters relative to the deforested area." line 502.

**The description of the age model (as suggested by reviewer 1).**

Added details and incorporated answers to reviewer 1 into the manuscript.

**2. Additionally, I would suggest the authors the following:**

**As the performance of the Ba/Ca ratio depends on compiling several cores within a single coral colony, could the materials section include more information on the distance between samples, and other differences between the sample location that can potentially explain why compiling this records results in a better correlation?**

There seem to be a confusion here, all cores come from different colonies which are located within both sites (EG and AG). Added details about distance and light conditions in the method sections. The distance between colonies is lower than 100 m, most within the measurement uncertainty of GPS readings.

**Can the discussion include a few lines on the potential to develop the Ba/Ca ratio as a quantitative tracer for either sea water Ba, sediment loading or even river discharge (or which parameters can be suggested)? Or would the authors suggest that this ratio will not be able to work as a quantitative proxy?**

Although we could only present one $Ba_{seawater}$ transect for the end of the dry season/start of wet season 2019 due to the Covid-19 pandemic preventing sampling in 2020 and 2021, we will collect the first wet season river to seawater transects in 2023.

Added "Our results highlight the potential of the $Ba/Ca_{coral}$ ratio as a quantitative tracer for dissolved barium in sediment loads of adjacent rivers. In cases where the relationship between Ba loading and river discharge is proportional (like ours, on our records' timescales) it also serves as tracer for river discharge. However, this study case also highlights the need for a long-term seawater monitoring program to measure the stability of the $Ba/Ca_{coral}$ and $Ba_{seawater}$ relationship by quantifying $Ba_{seawater}$ across wet and dry seasons for at least 1 year and at best for several years. Such data could cement $Ba/Ca_{coral}$ ratio as a quantitative tracer of $Ba_{seawater}$ thus making it able to quantitatively record Ba based sediment loading given a close enough proximity and current direction with a river mouth." line 506.